# Population Genetic Structure and Population History of the Biting Midge *Culicoides mahasarakhamense* (Diptera: Ceratopogonidae)

**DOI:** 10.3390/insects13080724

**Published:** 2022-08-13

**Authors:** Pairot Pramual, Panya Jomkumsing, Komgrit Wongpakam, Kotchaphon Vaisusuk, Wasupon Chatan, Bhuvadol Gomontean

**Affiliations:** 1Department of Biology, Faculty of Science, Mahasarakham University, Mahasalakan 44150, Thailand; 2Walai Rukhavej Botanical Research Institute, Mahasarakham University, Mahasalakan 44150, Thailand; 3Department of Veterinary Technology and Veterinary Nursing, Faculty of Agricultural Technology, Rajabhat Maha Sarakham University, Mahasalakan 44000, Thailand; 4Department of Veterinary Clinic, Faculty of Veterinary Sciences, Mahasarakham University, Mahasalakan 44000, Thailand

**Keywords:** *Culicoides*, insect vector, population expansion

## Abstract

**Simple Summary:**

Biting midges are important blood sucking insects. Many species are pests of humans and other animals, including economically important livestock. A new biting midge species, *Culicoides mahasarakhamense*, was recently described in Thailand. This species is a pest and a potential vector of blood protozoa transmitted to domestic chickens and other avian species. Knowledge of genetic diversity and genetic differentiation among populations of vector species is of paramount importance as it can be used to design effective control programs. However, this information is lacking for *C. mahasarakhamense*. In this study, we used mitochondrial cytochrome c oxidase I (COI) sequences as genetic markers to evaluate the level of genetic diversity and the genetic structure of *C. mahasarakhamense* from Thailand. The level of genetic diversity was relatively high, but most populations were not genetically different. However, one population from northern Thailand was found to be genetically highly different from others. Historical population isolation and geographic distance separation are possible explanations for the high level of genetic differentiation. This information will help in monitoring the spread of the parasitic disease agents carried by *C. mahasarakhamense* from different populations.

**Abstract:**

Biting midges of the genus *Culicoides* Latreille are significant pests and vectors of disease agents transmitted to humans and other animals. Understanding the genetic structure and diversity of these insects is crucial for effective control programs. This study examined the genetic diversity, genetic structure, and demographic history of *Culicoides mahasarakhamense*, a possible vector of avian haemosporidian parasites and *Leishmania martiniquensis*, in Thailand. The star-like shape of the median joining haplotype network, a unimodal mismatch distribution, and significant negative values for Tajima’s *D* and Fu’s *F*_S_ tests indicated that populations had undergone recent expansion. Population expansion time was estimated to be 2000–22,000 years ago. Population expansion may have been triggered by climatic amelioration from cold/dry to warm/humid conditions at the end of the last glaciations, resulting in the increased availability of host blood sources. Population pairwise *F*_ST_ revealed that most (87%) comparisons were not genetically different, most likely due to a shared recent history. The exception to the generally low level of genetic structuring is a population from the northern region that is genetically highly different from others. Population isolation in the past and the limitation of ongoing gene flows due to large geographic distance separation are possible explanations for genetic differentiation.

## 1. Introduction

Biting midges of the genus *Culicoides* Latreille (Diptera: Ceratopogonidae) are blood-sucking insect pests and vectors of viruses, protozoans, and nematodes transmitted to humans and other animals, including economically important livestock. The important disease-causing agents transmitted by biting midges include Oropouche fever in humans, bluetongue disease and epizootic hemorrhagic disease in ruminants, African horse sickness in equines, and leucocytozoonosis in domestic chickens [1,2,3].

An understanding of the genetic diversity, genetic structure, and population history of vector species provides important information required for effective disease control programs [4,5,6]. Knowledge of the level of genetic differentiation, and the rate and direction of gene flow can potentially be used for determining the spread of the disease agent. The level of genetic diversity and genetic structure can be used to determine the species status of the vector which could be related to the degree of competency (i.e., the ability of the vector to acquire pathogens and transmit them to a susceptible host) [6]. For example, different genetic lineages of the mosquito *Aedes albopictus* Skuse, originating from historical isolation and divergence, have different competencies for chikungunya virus [7]. Information gathered from a population genetic study could also be used to identify the sources of incursion by parasites and vector species [8,9].

Population genetic studies of *Culicoides* have generally found a low level of genetic differentiation within newly colonized areas (e.g., within a single country) but the genetic structure exists across the native range [8,9,10,11,12,13]. High dispersal capacity through wind assistance, long-distance dispersal followed by colonization and population expansion might explain this pattern of genetic structure [8,9]. For example, a population genetic study of *Culicoides imicola* Kieffer, a major vector species of the bluetongue virus (BTV) and African horse sickness viruses (AHSV) [14], revealed a high level of genetic structuring among populations within the native range in sub-Saharan Africa [12]. In contrast, low or no genetic structure exists in populations in recently colonized areas such as France [15] and Spain [10]. A similar situation has been observed in another vector species of BTV in Australia, *Culicoides*
*brevitarsis* Macfie. This biting midge species is native to Southeast Asia and has entered Australia recently. Because of the short history of *C. brevitarsis* in Australia, no genetic structure was detected [8].

*Culicoides mahasarakhamense* Pramual, Jomkumsing, Piraonapicha, Jumpato has recently been described from Maha Sarakham province, northeastern Thailand [16]. They have now been recorded in several locations in northeastern, central [17], and northern [18] regions of the country. Molecular blood meal analysis indicated that this midge is a chicken biter [19]. Molecular screening of the parasites found that *C. mahasarakhamense* was a potential vector of avian haemosporidian parasites, *Leucocytozoon* sp., that can cause Leucocytozoonosis, which is a disease found in many avian species including domestic chickens [1]. This biting midge species is also a potential vector of *Plasmodium juxtanucleare* and *P. gallinaceum* [17]. These plasmodium species can cause avian malaria which reduces the productivity of poultry [20]. Furthermore, a recent study found that *C. mahasarakhamense* was also a potential vector of the flagellated protozoan, *Leishmania martiniquensis* [18], the causative agent of Leishmaniasis. Therefore, this biting midge species is potentially a significant vector of disease agents transmitted to humans and other animals. Thus, understanding the genetic diversity, genetic structure, and population demographic history of *C. mahasarakhamense* will provide important baseline information for fully understanding the transmission ecology of these parasites that this biting midge species is a potential vector.

The parasitic disease agents found in *C. mahasarakhamense* are geographically restricted to particular locations. *Leucocytozoon* sp. was detected in one location from the northeast (Maha Sarakham (MK) province) and *Plasmodium juxtanucleare* and *P. gallinaceum* were found in another location (Roi Et (RE) province), also in the northeast of Thailand [17]. So far, the, flagellated protozoa, *Leishmania martiniquensis* has been found only in Lamphun province, northern Thailand [18]. Therefore, understanding the genetic diversity and genetic differentiation of *C. mahasarakhamense* will be useful for monitoring the spread of parasitic disease agents that might possibly be transmitted by this biting midge species. In this study we used mitochondrial cytochrome c oxidase subunit I (COI) sequences to infer the genetic diversity and genetic structure of *C. mahasarakhamense* in Thailand. In addition, we also inferred the population demographic history of this species to examine whether it experienced recent demographic expansion in response to climatic and environmental change during the Pleistocene, as has been found in other blood-sucking Diptera species such as black flies [21,22] and mosquitoes [23,24]. This information will also contribute to an understanding of the factors underlying the biodiversity of one of the most significant diversity hotspots.

## 2. Materials and Methods

### 2.1. Specimen Collections and Identification

Adult female specimens of *C. mahasarakhamense* were collected from eight sampling sites in Thailand between November 2020 and February 2021 (Table 1 and Figure 1). Information for five additional locations was retrieved from previous publications (four from [16,17] and one from [18]). All sampling locations were at or near cattle pens and/or chicken shelters where biting midges were abundant. Adult fly specimens were collected using a sweep-net, by sweeping around animals and randomly in the air close to the animal shelters. Specimen collections were conducted between 17:00 and 19:00 p.m. as biting midges are actively searching for a host blood meal during this time period. Specimens were preserved in 80% ethanol and stored at −20 °C until used. Specimens were identified using descriptions of Pramual et al. [16].

### 2.2. DNA Extraction, Polymerase Chain Reaction (PCR), and Sequencing

A total of 110 specimens were used for molecular study. DNA was extracted using the GF-1 Nucleic Acid Extraction Kit (Vivantis Technologies Sdn. Bhd, Shah Alam, Malaysia). Polymerase chain reaction (PCR) was used to amplify the COI gene barcoding region using the primers LCO1490 (5′-GGTCAACAAATCATAAAGATATTGG-3′) and HCO2198 (5′-TAAACTTCAGGGTGACCAAAAAATCA-3′) [25]. PCR reaction conditions followed those in [26]. PCR products were checked with 1% agarose gel electrophoresis and purified using a PureDireX PCR CleanUp & Gel Extraction Kit (Bio-Helix, Taiwan, China). DNA sequencing was performed at ATCG Company Limited (Thailand Science Park (TSP), Pathumthani, Thailand) using the same primers as for PCR. Among 110 specimens used for molecular study, only 86 were successful for PCR and sequencing.

### 2.3. Data Analysis

A total of 123 sequences with a sequence length of 647 bp were included in the data analyses; of these, 86 were obtained in the present study (GenBank accession nos. ON819766-ON819851) and the rest were retrieved from the NCBI GenBank database. Genetic relationships between mitochondrial DNA haplotypes were inferred using the median-joining (MJ) network [27] method in the software Network ver. 10.2.0.0 (https://www.fluxus-engineering.com (accessed on 2 February 2022)). Intraspecific genetic divergence was calculated based on the Kimura 2-parameter (K2P) model using TaxonDNA [28]. Genetic diversity indices based on haplotype diversity (h) and nucleotide diversity (π) were calculated using the K2P model in Arlequin ver. 3.5 [29]. Genetic differentiation between populations based on pairwise *F*_ST_ was calculated in Arlequin using 1023 permutations for each statistical test. The sequential Bonferroni correction was used to adjust the significance level for multiple tests. To avoid bias as a result of a small sampling size, populations with sample sizes of less than three were not included in the *F*_ST_ analysis. The relationship between the level of genetic differentiation (based on *F*_ST_ values) and geographic distance (km) was used to test the isolation-by-distance (IBD) model using Mantel’s test [30]. The IBD model was tested using IBD ver. 1.52 [31] with 1000 randomizations.

To test whether the population had undergone recent demographic expansion, mismatch distribution analysis was conducted in Arlequin. A unimodal mismatch distribution provides a signal of population expansion [32]. The sum-of-squares deviation (SSD) and Harpending’s raggedness index [33] were used to test the deviation from sudden or spatial expansion models, respectively. The mismatch distribution, SSD, and Harpending’s raggedness index were estimated in Arlequin. If the mismatch distribution analysis indicated a signal of population expansion, the expansion times were estimated using the equation τ = 2ut (where u = m_T_μ, m_T_ is the length of the nucleotide sequences (647 bp) under study, μ is the mutation rate per nucleotide, and t is the generation time) [32]. The divergence rate of 3.54% per million years for insect COI gene [34] was used. The generation time of *C. mahasarakhamense* is unknown. Therefore, we assumed that it was similar to the closely related species, *C. arakawae* (Arakawa) that has a generation time of approximately one month [35] which was used for the population expansion time calculation.

## 3. Results

### 3.1. Genetic Diversity of Culicoides Mahasarakhamense

A total of 56 haplotypes were identified among 123 COI sequences. The overall haplotype diversity was 0.8517. Within each population, the haplotype diversity ranged between 0 and 1.0000 (Figure 2). The overall nucleotide diversity was 0.0040 and ranged between 0 and 0.0075 in each population (Figure 2). Intraspecific genetic divergence based on the K2P model ranged between 0 and 3.0465% with an average of 0.4017%.

### 3.2. Mitochondrial Genealogy and Population Genetic Structure

The MJ network (Figure 3) of 123 COI sequences revealed neither major genetic breaks nor geographic associations of the haplotypes. Overall, the MJ network was a star-like shape in which the central haplotype had the highest frequency (47 specimens from the northeast and central regions shared this haplotype). Almost all haplotypes connected to the central haplotype by short branch lengths, with the exception of three long divergent haplotypes. These genetically divergent haplotypes were all from the northeastern region of Thailand (SK and NL populations). Two haplotypes from the SK population were 1.05–2.43% divergent from the main lineage. A haplotype from NL was 1.50–2.42% divergent from the main lineage.

Population genetic structure based on population pairwise *F*_ST_ analysis revealed that most (48 from 55) comparisons expressed no significant genetic differentiation. Seven comparisons were significantly statistically different. Five of these were comparisons between LP from the northern region with other populations. The insignificant comparisons between LP and five other populations (MK1, MK2, RE, NL, LO) were likely due to small sample sizes (*n* ≤ 10) of the latter five populations as both showed high (>0.301) *F*_ST_ values (Figure 4). Two other comparisons that were statistically significantly different were between SR and RE and NL, all from the northeastern region. Isolation-by-distance analysis revealed no significant (r^2^ = 0.2706, *p* = 0.0980) relationship between genetic and geographic distance.

### 3.3. Population History

Mismatch distribution analysis produced a unimodal graph (Figure 5), but the SSD was statistically significantly different from the simulation under the sudden expansion model (SSD = 0.1135, *p* < 0.0001). However, when the observed data were fitted with the spatial expansion model, SSD (0.0028, *p* = 0.5500), and Harpending’s Raggedness index (0.0302, *p* = 0.6500), there was no significant difference between the observed and simulated data. Tajima’s D (−2.5734, *p* < 0.0001) and Fu’s *F*_S_ (−26.6812, *p* < 0.0001) were also highly significantly negative, supporting the population expansion hypothesis. Population expansion time was estimated to be 2884 years ago (95% C.I. 2011–22,199 years ago).

## 4. Discussion

The K2P intraspecific genetic divergence based on COI sequences of *C. mahasarakhamense* found in this study (3.05%) was higher than previously reported (max. 2.03%) [16]. This is not unexpected given the larger sample size (123 vs. 30) and wider geographic coverage of the sampling locations (north, northeast, central vs. northeast). The relatively high intraspecific genetic divergence in *C. mahasarakhamense* found in this study is due to the presence of three divergent haplotypes identified here. If these three divergent haplotypes were omitted, the maximum K2P intraspecific genetic divergence would only have been 1.66%. However, the level of genetic divergence of these haplotypes (max. 2.43%) compared to the main lineage occurred within the expected range of intraspecific genetic variations of *Culicoides* species [17,36,37,38,39]. Although the level of genetic divergence fell within the range of the within-species genetic divergence of *Culicoides* species, the relatively high level of genetic divergence of these three haplotypes compared to the rest could be indicative of the presence of cryptic species or an early stage of divergence. Further study with more specimens included from populations that possessed the divergent haplotypes is needed to test this hypothesis.

Population genetic structure analysis revealed that most comparisons (87%) were not genetically significantly different. The exceptions to the general overall low level of genetic structuring were comparisons between a population from the north (LP) with others, as five from ten comparisons were genetically significantly different. Although the *F*_ST_ values between LP and five other populations (MK1, MK2, RE, NL, LO) were not statistically significant, the values were relatively high (>0.26) compared to other comparisons (<0.06). Therefore, no statistical significance between LP and these populations was most likely because of the small sample sizes (*n* ≤ 10) rather than the high gene flow between these populations. High *F*_ST_ values between LP and the other populations was due to the presence of two haplotypes that were genetically relatively highly different from others. These haplotypes are possibly relicts of ancestral polymorphism. Three other specimens of the LP population were connected directly to the central haplotype with only a single mutation step. Among these, one specimen shared a haplotype with the UB population in the northeastern region from which it is geographically separated by >700 km. Haplotype sharing with populations in the northeast and two others that are genetically closely related to the most common haplotype indicated that there has been gene flow between LP and other populations, in contradiction with the high *F*_ST_ values. The LP population is geographically separate from the remaining populations included in this study by >420 km. This geographic distance is far beyond the active flight range of biting midges which have a maximum of only 6 km [40]. There are two possible explanations for long-distance gene flow between these populations. First, the gene flow is mediated by wind-assisted long-distance dispersal. This mechanism was postulated as a factor responsible for the incursion of some vector species such as *C. imicola* [9], *C. obsoletus* [13], and *C. brevitarsis* [8] across long geographic distances. However, long-distance dispersal is not common because the successful colonization of new areas requires the presence of many suitable environmental conditions such as climate, ecology of the new habitat, and availability of hosts [41]. In addition, the long distance dispersal of biting midges on land by wind is usually much shorter (<85 km) than over sea (up to 700 km) [41,42]. Furthermore, there are two major monsoon winds in Thailand. The northwest monsoon begins in May and flows from the southwest to the northeast bringing moisture from Indian Ocean to the country and generating the rainy season. The northeast monsoon begins in October and flows from the northeast to the southwest bringing cold and dry air from the higher latitude in China to the country generating the cold season [43]. *Culicoides* is highly abundant during the rainy season (southwest monsoon). If wind-assisted long-distance dispersal actually occurs, it is more likely to be between populations in the southwest to the northeast regions. However, the LP and UB sites are located along a northwest to southeast directional axis; thus, the southwest monsoon wind is unlikely to have assisted long-distance dispersal between these populations. The second and more likely explanation for the gene flow between LP and UB is that it is mediated by human agency. Biting midge species can be transported with livestock [44] and livestock transportation (e.g., of chickens) is very common in Thailand. It might be possible that *C. mahasarakhamense* can be transported with livestock from UB to LP or vice versa resulting in gene flow between these populations. However, this long-distance gene flow will be very rare given that there is a large *F*_ST_ value between LP and other populations.

In contrast to the situation of LP, the RB population is located in the central region with >480 km between it and the other populations included in this study but with low *F*_ST_ values (<0.081) and only LP is significantly different from the others. This result indicates a high gene flow between RB and other populations. However, the genetic connectivity between RB and other populations in the northeast is more likely to be related to historical movement as indicate by the interior position of the haplotypes of this population in the MJ network [45]. Mismatch distribution analysis revealed that the *C. mahasarakhamense* species had undergone spatial population expansion dating back to approximately 2000–22,000 years ago. This population expansion is possibly related to the recovery from cold and dry conditions at the end of the Pleistocene with ensuing warm and humid conditions of the warming period. Although *Culicoides* does not strictly require free water for reproduction, they do need a high moisture content substrate for development of immature stages [3]. The adults would also find humid conditions advantageous because the dry conditions could cause serious dehydration problems due to their small size combined with a relatively large surface area. The historical, long-distance movement between RB and populations in the northeast of Thailand could be a result of wind assistance by the northwest monsoon wind. As explained above, the southwest monsoon wind flows from southwest to the northeast and could bring biting midges from RB to the northeastern region.

Climatic amelioration will have produced a more favorable environment and the historical population expansion of *C. mahasarakhamense* could also be associated with the increasing population density of the host blood source, the domestic chickens. A previous study based on the molecular identification of host blood indicated that the *C. mahasarakhamense* species are chicken biters [16,19]. A high density of this species was found in a chicken shelter [16]. A recent study found that the domestic chicken was first domesticated in northeastern Thailand around 3200–3600 years ago [46]. Thus, the increasing host density could potentially have led to the population expansion of *C. mahasarakhamense*. A similar situation has been reported in a mammalophilic black fly species, *Simulium nodosum* Puri in Thailand [22]. The population expansion of this black fly species was dated back to 2600–5200 years ago which related to the time of agricultural proliferation leading to increased domestic animal and human population densities [22].

## 5. Conclusions

In conclusion, the population genetic structure and population history analyses revealed that both ongoing and historical factors have contributed to the genetic diversity and genetic structure of *C. mahasarakhamense*. A population from the north (LP) that was reported as a possible vector of *Leishmania martiniquensis* [20] is genetically distinct from others. Geographic distance is most likely a limiting factor for gene flow. However, there was a geographically large sampling gap between the northern population and others included in this study. Therefore, it will be useful for further study to examine populations located between the LP and other populations included in this study. This will enable the testing of the hypothesis that geographic distance is a barrier to gene flow for *C. mahasarakhamense*. This information could be useful for monitoring the spread of disease agents potentially transmitted by this biting midge species.

## Figures and Tables

**Figure 1 insects-13-00724-f001:**
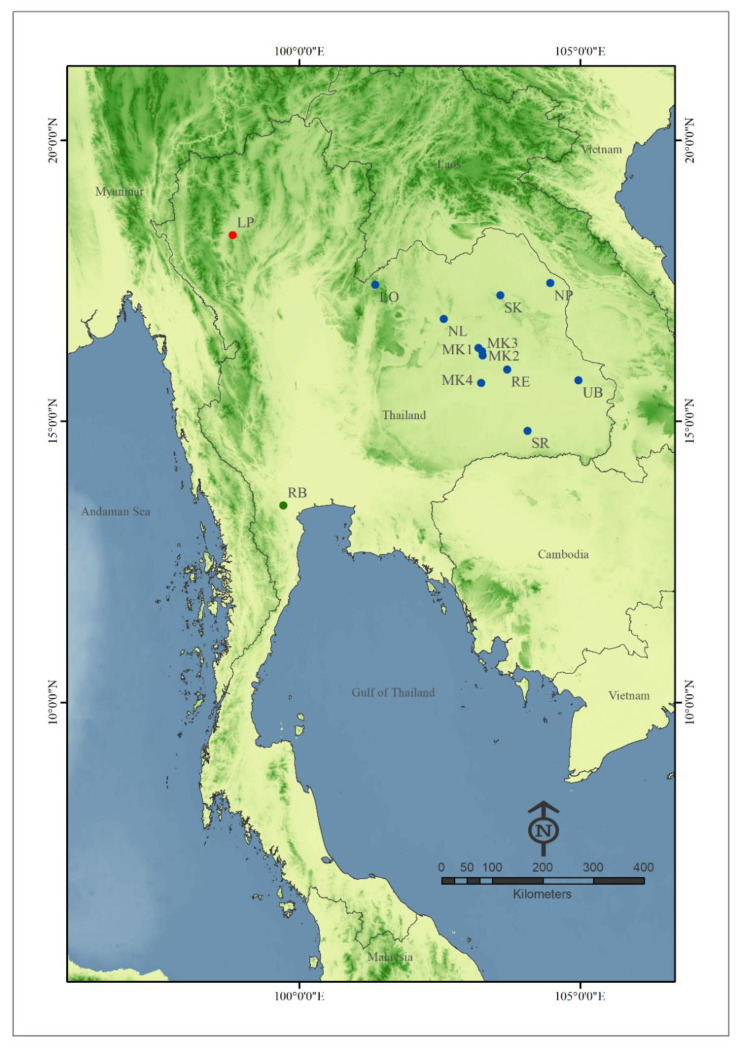
Map of Thailand (modified from http://mitrearth.org (accessed on 1 August 2022)) indicating the 13 sampling locations of *Culicoides mahasarakhamense* used in this study. Details of sampling locations are included in Table 1. Locality symbols are labeled according to geographic region: red, north; blue, northeast; green, central.

**Figure 2 insects-13-00724-f002:**
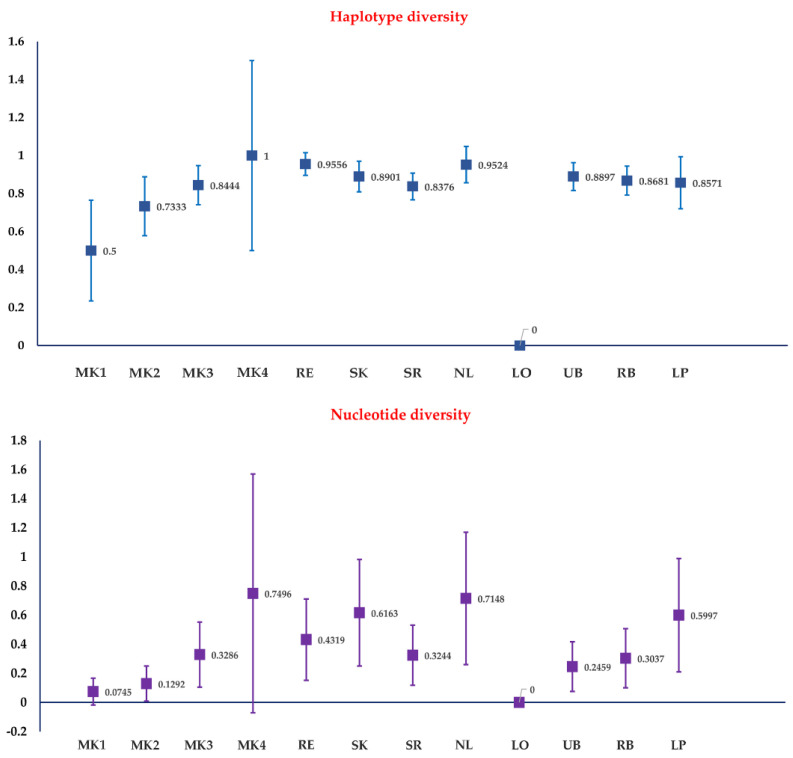
Haplotype diversity and percentage nucleotide diversity (±standard deviation) of 12 populations of *Culicoides mahasarakhamense* in Thailand.

**Figure 3 insects-13-00724-f003:**
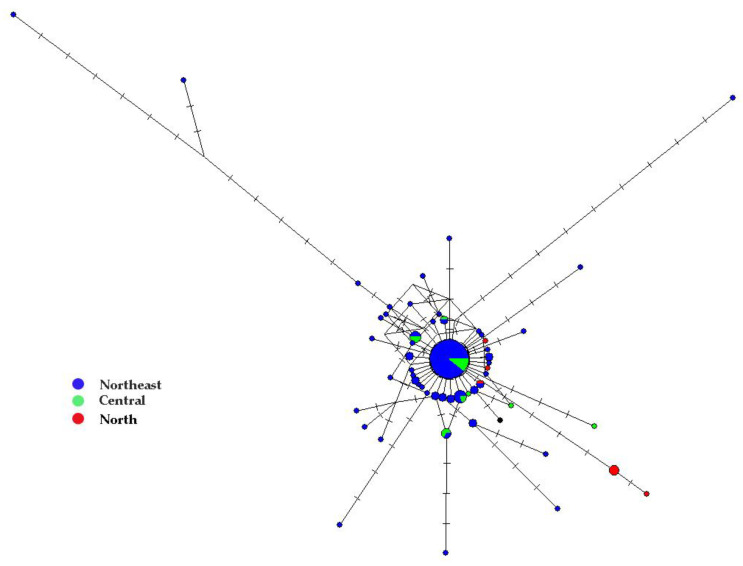
Median joining network of 123 mitochondrial cytochrome c oxidase I (COI) sequences of *Culicoides mahasarakhamense* in Thailand. Each circle represents a haplotype and sizes are relative to the number of individuals sharing such haplotypes. Haplotypes are labelled according to the geographic region as in Figure 1. Crossbars indicate mutation steps.

**Figure 4 insects-13-00724-f004:**
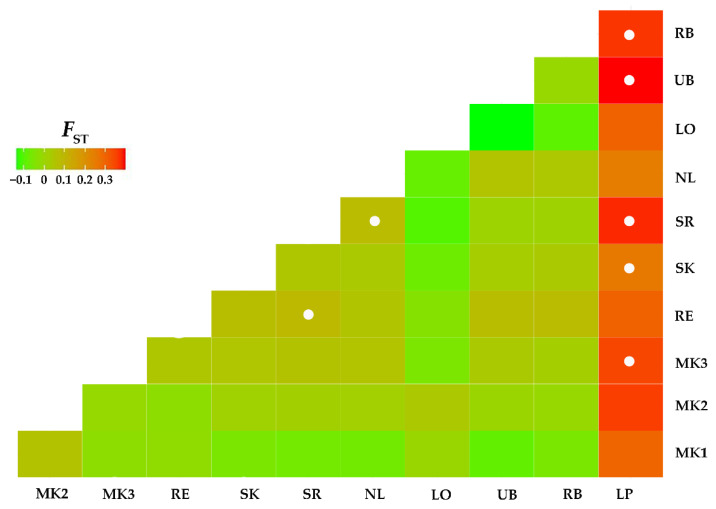
Heatmap for the pairwise *F*_ST_ values between 11 populations of *Culicoides mahasarakhamense* in Thailand calculated based on COI sequences using the Kimura 2-parameter model. White circles indicate statistical significance after Bonferroni correction. Details of populations are provided in Table 1.

**Figure 5 insects-13-00724-f005:**
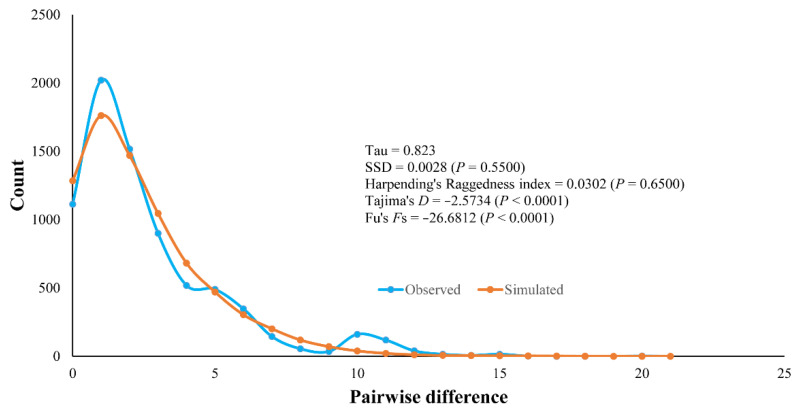
Mismatch distribution based on 123 mitochondrial cytochrome c oxidase I (COI) sequences of *Culicoides mahasarakhamense* in Thailand. Values of sum-of-square deviation (SSD), Harpending’s Raggedness index, Tajima’s *D*, and Fu’s *F*_S_ tests are given.

**Table 1 insects-13-00724-t001:** Sampling locations of *Culicoides mahasarakhamense* specimens in Thailand used in this study.

Location (Code)	Sampling Site Type	Region	N (*h*_n_)	Latitude/Longitude	Elevation (m)	Collection Date
1. Ban Wai, Kantharawichai District, Maha Sarakham (MK1)	CS	Northeast	4 (2) ^1^	16.3075° N/103.1891° E	150	20 January 2021
2. Lerng Jan Reservior, Mueang Maha Sarakham District, Maha Sarakham Province (MK2)	CP	Northeast	6 (3) ^1^	16.1730° N/103.2622° E	140	28 January 2020
3. Mahasarakham University, Kantharawichai District, Maha Sarakham Province (MK3)	CP	Northeast	10 (6) ^1^	16.2488° N/103.2505° E	150	25 February 2019
4. Nadun District, Maha Sarakham Province (MK4)	CP/CS	Northeast	2 (2)	15.6830° N/103.2358° E	140	20 February 2021
5. Ban Nong Bon, Mueang Roi Et District, Roi Et Province (RE)	CP/CS	Northeast	10 (8) ^1^	15.9241° N/103.6980° E	130	9 February 2020
6. Waritchaphum District, Sakon Nakhon Province (SK)	CP/CS	Northeast	14 (10)	17.2422° N/103.5744° E	220	27 March 2021
7. Phon Sawan District, Nakhon Phanom Province (NP)	CP	Northeast	1 (1)	17.4616° N/104.4658° E	150	20 November 2020
8. Prangku District, Sisaket Province (SR)	CP/CS	Northeast	27 (15)	14.8305° N/104.0605° E	140	6 March 2021
9. Non Sang District, Nongbua Lampu Province (NL)	CP/CS	Northeast	7 (6)	16.8233° N/102.5688° E	180	27 February 2021
10. Phu Ruea District, Loei Province (LO)	CS	Northeast	4 (4)	17.4308° N/101.3500° E	650	20 January 2021
11. Kud Khaopun District, Ubon Ratchathani Province (UB)	CP/CS	Northeast	17 (12)	15.7330° N/104.9680° E	130	13 January 2021
12. Paktho District, Ratchaburi Province (RB)	CP/CS	Central	14 (8)	13.5058° N/99.7125° E	5	1 November 2021
13. Banhong District, Lamphun Province (LP)	N/A	North	7 (5) ^2^	18.3169° N/98.8141° E	300	September 2019–January 2020
Total			123			

CS, chicken shelter; CP, cattle pens. *h*_n_, number of haplotype. ^1^ Data from Pramual et al. [16] and Jomkumsing et al. [17]; ^2^ Data from Sunantaraporn et al. [18].

## Data Availability

The data generated during the study have already been reported in the manuscript.

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
