# Peer review of "Population Genetic Structure and Population History of the Biting Midge *Culicoides mahasarakhamense* (Diptera: Ceratopogonidae)"

_insects, 2022, doi:10.3390/insects13080724_

Round 1

Reviewer 1 Report

Pairot Pramual et al. In Insects-1834788 MS.

In Insects-1834788 MS “Population genetic structure and population history of the biting midge Culicoides mahasarakhamense (Diptera: Ceratopogonidae)” Pairot Pramual and co-authors provide the first description of the genetic structure and diversity of the recently recognized Culicoides mahasarakhamense Pramual, Jomkumsing, Piraonapicha, Jumpato in Thailand. Field based studies in this country suggest that C. mahasarakhamense may be a biological vector involved in the transmission of avian hemoparasites (Leucocytozoon sp and Plasmodium spp.) impacting in backyard chickens and poultry industry while also participates in the transmission of Leishmania martiniquensis a zoonotic neglected tropical disease. Previous research by the authors about the COI sequence diversity in Culicoides in Thailand provided baseline information (moderate intraspecific genetic divergence) for exploring the genetic diversity in C. mahasarakhamense. In accordance with the current understanding of the genetic structure for other Culicoides species with wide distribution range and subject to geographical barriers, like Culicoides imicola Kieffer and Culicoides brevitarsis Macfie, in this manuscript the authors attempted to do the same. Although, this study focused on a narrow geographical range and did not provide any theorical background that justify that C. mahasarakhamense population structure could be shaped by the interaction between environmental barriers and the effect of wind assisted dispersal.

Pairot Pramual et al. in Insects-1834788 MS is out of the scope of this special collection about Vector-Borne Diseases in a Changing World since the results presented does not contribute in any of the potential dimensions that the global change may impact in Culicoides-borne diseases. In addition, there are majors concerns that may justify re-submit this research as a brief-communication:

1-    the MS lacks a robust theorical justification about the relevance for studying the genetic structure of C. mahasarakhamense into regards of parasite transmission. In addition, information regarding C. mahasarakhamense geographical distribution, potential physical barriers that could constrain its distribution and the main wind pattern that could favor midges dispersal are also lacking. This must be outlined in the introduction section.

2-    The study designs could be inappropriate to the scope of this study and needs to be improved with a sampling strategy that cover the C. mahasarakhamense distribution range at least in Thailand. The Thailand peninsular had not been represented within the sample, the central and north regions samples came from one sample location each and the northeast area clumped 11locations. It concerns the unbalanced distribution between collection sites, the collection method and the small number of specimens collected in each site. For example, Table 1 shows that 11 out of 13 sites and 82% of the specimens collected belonged to NE Thailand.

3-    Is it COI data enough when genetic structure and diversity is studied or should be complemented with microsatellites or others nuclear genes? Nuclear genes and microsatellites could be a more appropriate genetic marker for exploring and uncover population structure at the imposed geographical range of this study (~500 km). References cited in this MS (Onyango et al. 2015a,b) have accounted nuclear markers. If the authors considered nuclear markers are not required to accomplish the study aim, they may provide a clear justification.

4-    A statistical model framework that take in consideration the spatial aggregation between the population surveyed and the unbalanced distribution between Thailand regions sampled it is also necessary.

Specific comments:

Lines 46-48: The first two first sentences should be reduced in one.”… are blood sucking insects pest to humans and other animals including...”

Line 48: The use of “parasitic disease” sounds redundant in this context. "..vectors of virus, protozoans..” seem enough.

Lines 50-52: Please, rephrase this sentence. Diseases are not transmitted, they are developed in the hosts as consequence of the interaction between the host, parasite or pathogen agents and the environment. Therefore, parasites are the ones biologically transmitted.

Line 56: An example is necessary to account for the potential effect of genetic differentiation on vectorial competence and/or capacity. Also, Martinez de la Puente et al. 2011 did not investigate or discuss within Culicoides species competence ability. Instead of that, they discussed the potential effect of the variation in host association for vectorial capacity. Competence and Capacity should be shortly defined in this section of the introduction and the authors provide examples on how genetic differentiation between populations could affect parasites (in the broad ecological sense) transmission.

Line 63: Culicoides may disperse but not by migration. Long range passive or assisted dispersal disagree with migration means. I suggest Service 1997 (Journal of medical entomology, 34(6), 579-588) reading.

Line 65: The complete scientic name of species mut be used when is first mentioned in the manuscript (i.e., Culicoides imicola Kieffer).

Lines 65-66: “bluetongue virus and African horse sickness” need to be changed to “blue tongue and African horse sickness viruses”. Authors may abbreviate virus names after first mentioned as BTV and AHSV.

Line 75: Considere the use of  “Molecular blood meal analysis” when “Molecular analysis of host blood meals” is used.

Line 75-84: Even though preliminary blood meal data and the records of specimens naturally infected are a good starting in the putative incrimination of blood suckling arthropods as vector. The vectorial role of C. mahasarakhamense should be stated as "potential" until more observational and experimental studies will be published. Although, it is well recognized the challenge of testing vector competence in Culicoides, the information provided only support that C. mahasarakhamense is susceptible to infection.

Lines 85-95: This paragraph have to highlight the aims, hypothesis and relevance of the study. Therefore, It have to be stated the exploratory aim of this study and a clear messages about that the genetic structure and diversity plus the potential population structure were explored. Please, special attention should be taken to avoid odd arguments like stated in the first sentence. Provide arguments that justify exploring the genetic structure/diversity of C. mahasarakhamense considering parasite transmission. Also, background information about the species and barriers that justify this research. This is all about what i would look in the introduction sections

Line 91: This sentence did not describe properly the current know distribution of C. mahasarakhamense.

Line 100: In the Table 1 information regarding the main features of the collections sites or ecotypes need to be described. I suggest that sites' geopositions be expressed as Lat/Long decimal degrees.

Line 103: was abundant instead of “were”.

Line 130: The spatial dependence within the NE regions is concerning, so initially fitting a mixed effect model will be appropriate by considering how samples were clustered within regions. Also, it is necessary to explicit if linear o geodesic geographic distance was used.

Line 132: The analysis used to test IBD (isolation by distance) have to be described (Mantel test, Procrustes, RDA).

Line 137: Change “models.” By “models, respectively.”.

Line 150: Figure 1. The map can be improved by the addition of a digital elevation layer, a sea/cost color layer and the labels for the Gulf of Thailand and the Andaman Sea. This will help readers to have a quick idea about biogeographical barriers.

Line 160: Extracting the message from haplotype and nucleotide diversity from Table 1 is odd. These results can be summarized with two contiguous figures as error-bars barplots. The nucleotide diversity values should be transformed.

Lines 189-191: Then demographic expansion was not supported by the data and the patterns of historical demography should be inferred in the future by Bayesian Skyline Plot analysis to test this population expansion hypothesis.

Line 197: Significant threshold for Fst is 0.02. Have been these p values been corrected by multiple hypothesis testing? By which method? A heat plot could be a better representation of the Fs pairwise matrix.

Line 263- 264: Jomkumsing et al. 2021 (Reference #17) has not proper data to indicate C. mahasarakhamense is a chicken biter (as mean chicken host preference) because sampling biases by collecting blood engorged specimens of Culicoides arawakae Arakawa near cattle pens and/or chicken shelters. In addition, Jomkumsing et al. 2021 reported the bloodmeal analysis of 25 C. arawakae, it is not clear if all the specimens actually belong to C. mahasarakhamense. Pramual et al. 2021 (Reference #18) did not perform bloodmeal analysis that support C. mahasarakhamense is a “chicken biter”.

Author Response

Comments and Suggestions for Authors: Pairot Pramual et al. In Insects-1834788 MS.

In Insects-1834788 MS “Population genetic structure and population history of the biting midge Culicoides mahasarakhamense (Diptera: Ceratopogonidae)” Pairot Pramual and co-authors provide the first description of the genetic structure and diversity of the recently recognized Culicoides mahasarakhamense Pramual, Jomkumsing, Piraonapicha, Jumpato in Thailand. Field based studies in this country suggest that C. mahasarakhamense may be a biological vector involved in the transmission of avian hemoparasites (Leucocytozoon sp and Plasmodium spp.) impacting in backyard chickens and poultry industry while also participates in the transmission of Leishmania martiniquensis a zoonotic neglected tropical disease. Previous research by the authors about the COI sequence diversity in Culicoides in Thailand provided baseline information (moderate intraspecific genetic divergence) for exploring the genetic diversity in C. mahasarakhamense. In accordance with the current understanding of the genetic structure for other Culicoides species with wide distribution range and subject to geographical barriers, like Culicoides imicola Kieffer and Culicoides brevitarsis Macfie, in this manuscript the authors attempted to do the same. Although, this study focused on a narrow geographical range and did not provide any theorical background that justify that C. mahasarakhamense population structure could be shaped by the interaction between environmental barriers and the effect of wind assisted dispersal.

Pairot Pramual et al. in Insects-1834788 MS is out of the scope of this special collection about Vector-Borne Diseases in a Changing World since the results presented does not contribute in any of the potential dimensions that the global change may impact in Culicoides-borne diseases.

-     Thank you very much for your valuable comments and suggestions which are much appreciated and very helpful for improvement of the manuscript. Our results indicated that genetic structure, diversity and population demographic history of Culicoides mahasarakhamense, a potential vector of haemosporidian blood protozoa and Leishmania martiniquensis, were influenced by both recurrent (e.g. human mediated, long distance dispersal) and historical (e.g. Pleistocene climatic and environmental change, wind assisted long distance dispersal) factors. Both recurrent and historical events that shaping genetic structure and diversity of C. mahasarakhamense are dynamic factors intimately related to the “changing world”. Therefore, we believe that our manuscript is within the scope of Insects journal under the topical collection "Vector-Borne Diseases in a Changing World".

In addition, there are majors concerns that may justify re-submit this research as a brief-communication:

1-    the MS lacks a robust theorical justification about the relevance for studying the genetic structure of C. mahasarakhamense into regards of parasite transmission. In addition, information regarding C. mahasarakhamense geographical distribution, potential physical barriers that could constrain its distribution and the main wind pattern that could favor midges dispersal are also lacking. This must be outlined in the introduction section.

      - Thank you for your comment and suggestion, we have modified the Introduction section and provided a theoretical justification for studying the genetic structure of C. mahasarakhamense. More details of geographic distribution are also provided in the Introduction.

2-    The study designs could be inappropriate to the scope of this study and needs to be improved with a sampling strategy that cover the C. mahasarakhamense distribution range at least in Thailand. The Thailand peninsular had not been represented within the sample, the central and north regions samples came from one sample location each and the northeast area clumped 11locations. It concerns the unbalanced distribution between collection sites, the collection method and the small number of specimens collected in each site. For example, Table 1 shows that 11 out of 13 sites and 82% of the specimens collected belonged to NE Thailand.

      - We agree that the sampling sites and number of specimen/site do not adequately cover the distribution range of C. mahasarakhamense. We realized this limitation and have mentioned this in the conclusion section.

3-    Is it COI data enough when genetic structure and diversity is studied or should be complemented with microsatellites or others nuclear genes? Nuclear genes and microsatellites could be a more appropriate genetic marker for exploring and uncover population structure at the imposed geographical range of this study (~500 km). References cited in this MS (Onyango et al. 2015a,b) have accounted nuclear markers. If the authors considered nuclear markers are not required to accomplish the study aim, they may provide a clear justification.

      - We agree that additional genetics marker such as microsatellites or other nuclear genes would be helpful. However, the level of genetic diversity, with maximum K2P intraspecific genetic divergence of 3.0465% is relatively high compare to other Culicoides species. This indicated that COI sequences used as genetic markers in the present study have sufficient variation for the inference of genetic diversity and genetic structure.

4-    A statistical model framework that take in consideration the spatial aggregation between the population surveyed and the unbalanced distribution between Thailand regions sampled it is also necessary.

 - - Thank you for your recommendation. We have used the linear geographic distance. We agree that the sampling sites are almost from the NE and stated this limitation in the conclusion section. 

Specific comments:

Lines 46-48: The first two first sentences should be reduced in one.”… are blood sucking insects pest to humans and other animals including...”

- Thank you for your suggestion, we have modified this accordingly.

Line 48: The use of “parasitic disease” sounds redundant in this context. "..vectors of virus, protozoans..” seem enough.

 - Thank you for your suggestion, we have modified this accordingly.

Lines 50-52: Please, rephrase this sentence. Diseases are not transmitted, they are developed in the hosts as consequence of the interaction between the host, parasite or pathogen agents and the environment. Therefore, parasites are the ones biologically transmitted.

-  Thank you for your suggestion, we have modified this accordingly.

Line 56: An example is necessary to account for the potential effect of genetic differentiation on vectorial competence and/or capacity. Also, Martinez de la Puente et al. 2011 did not investigate or discuss within Culicoides species competence ability. Instead of that, they discussed the potential effect of the variation in host association for vectorial capacity. Competence and Capacity should be shortly defined in this section of the introduction and the authors provide examples on how genetic differentiation between populations could affect parasites (in the broad ecological sense) transmission.

- An example of potential effect of genetic differentiation on vectorial competence and the definition of vectorial competency were provided in the second paragraph of the Introduction.

Line 63: Culicoides may disperse but not by migration. Long range passive or assisted dispersal disagree with migration means. I suggest Service 1997 (Journal of medical entomology, 34(6), 579-588) reading.

- Thank you for this suggestion. We have changed “migration” to “dispersal” in this sentence.

Line 65: The complete scientic name of species mut be used when is first mentioned in the manuscript (i.e., Culicoides imicola Kieffer).

- Corrected

Lines 65-66: “bluetongue virus and African horse sickness” need to be changed to “blue tongue and African horse sickness viruses”. Authors may abbreviate virus names after first mentioned as BTV and AHSV.

- Thank you for this suggestion, we have modified the manuscript accordingly.

Line 75: Considere the use of “Molecular blood meal analysis” when “Molecular analysis of host blood meals” is used.

- Thank you for suggestion, we have modified the manuscript accordingly.

Line 75-84: Even though preliminary blood meal data and the records of specimens naturally infected are a good starting in the putative incrimination of blood suckling arthropods as vector. The vectorial role of C. mahasarakhamense should be stated as "potential" until more observational and experimental studies will be published. Although, it is well recognized the challenge of testing vector competence in Culicoides, the information provided only support that C. mahasarakhamense is susceptible to infection.

 - Thank you for this suggestion, we have modified the manuscript accordingly.

Lines 85-95: This paragraph have to highlight the aims, hypothesis and relevance of the study. Therefore, It have to be stated the exploratory aim of this study and a clear messages about that the genetic structure and diversity plus the potential population structure were explored. Please, special attention should be taken to avoid odd arguments like stated in the first sentence. Provide arguments that justify exploring the genetic structure/diversity of C. mahasarakhamense considering parasite transmission. Also, background information about the species and barriers that justify this research. This is all about what i would look in the introduction sections

- Thank you very much for your suggestions. We have modified this section accordingly.

Line 91: This sentence did not describe properly the current know distribution of C. mahasarakhamense.

- We have modified this sentence. More details of the current known geographic distribution of C. mahasarakhamense was provided.

Line 100: In the Table 1 information regarding the main features of the collections sites or ecotypes need to be described. I suggest that sites' geopositions be expressed as Lat/Long decimal degrees.

- Table 1 has been modified, types of the sampling site (i.e. chicken shelter or cattle pens or both) were added. Geographic positions were change to decimal degree format.

Line 103: was abundant instead of “were”.

- Corrected

Line 130: The spatial dependence within the NE regions is concerning, so initially fitting a mixed effect model will be appropriate by considering how samples were clustered within regions. Also, it is necessary to explicit if linear o geodesic geographic distance was used.

- Thank you for your recommendation. We have used linear geographic distance. We agree that the sampling sites are almost from the NE and stated this limitation in the conclusion section. 

Line 132: The analysis used to test IBD (isolation by distance) have to be described (Mantel test, Procrustes, RDA).

- We used a Mantel test to test IBD. This information was added.

Line 137: Change “models.” By “models, respectively.”.

- Corrected.

Line 150: Figure 1. The map can be improved by the addition of a digital elevation layer, a sea/cost color layer and the labels for the Gulf of Thailand and the Andaman Sea. This will help readers to have a quick idea about biogeographical barriers.

- Figure 1 was modified following reviewer recommendation.

Line 160: Extracting the message from haplotype and nucleotide diversity from Table 1 is odd. These results can be summarized with two contiguous figures as error-bars barplots. The nucleotide diversity values should be transformed.

- Thank you very much for your suggestions. We have removed Table 2 and was replaced it with Figure 2 that present error-bar plots for haplotype and nucleotide diversity.

Lines 189-191: Then demographic expansion was not supported by the data and the patterns of historical demography should be inferred in the future by Bayesian Skyline Plot analysis to test this population expansion hypothesis.

- Thank you for your recommendation. However, we consider that mismatch distribution analysis along with associated statistics (i.e. Harpending’s raggedness index, sum-of-squares deviations, Tajima’s D and Fu’s Fs tests) are sufficient for testing the hypothesis of population expansion. There are several recent publications that used the same analytical method with the present manuscript such as:

- Li, X., Wu, S., Xu, Y., Liu, Y., & Wang, J. (2022). Population Genetic Structure of Chlorops oryzae (Diptera, Chloropidae) in China. Insects, 13(4), 327.

- Mohamed, W. M. A., Moustafa, M. A. M., Thu, M. J., Kakisaka, K., Chatanga, E., Ogata, S., ... & Nakao, R. Comparative mitogenomics elucidates the population genetic structure of Amblyomma testudinarium in Japan and a closely related Amblyomma species in Myanmar. Evolutionary Applications. https://doi.org/10.1111/eva.13426

Line 197: Significant threshold for Fst is 0.02. Have been these p values been corrected by multiple hypothesis testing? By which method? A heat plot could be a better representation of the Fs pairwise matrix.

 - Thank you for your recommendation. The FST p-values have been adjusted using Bonferroni correction in the revised manuscript. A heatmap plot is now used to present FST values between populations instead of a table of pairwise matrix.

Line 263- 264: Jomkumsing et al. 2021 (Reference #17) has not proper data to indicate C. mahasarakhamense is a chicken biter (as mean chicken host preference) because sampling biases by collecting blood engorged specimens of Culicoides arawakae Arakawa near cattle pens and/or chicken shelters. In addition, Jomkumsing et al. 2021 reported the bloodmeal analysis of 25 C. arawakae, it is not clear if all the specimens actually belong to C. mahasarakhamense. Pramual et al. 2021 (Reference #18) did not perform bloodmeal analysis that support C. mahasarakhamense is a “chicken biter”.

- Jomkumsing et al. (2021) reported that molecular analysis of blood meals obtained from 15 blood engorged females of “C. arakawae” were from chicken blood. Therefore, they concluded that C. arakawae is chicken biter. Later study by Pramual et al. (2021) indicated that some specimens previously recognized as “C. arakawae” by Jomkumsing et al. (2021) are different species. Thus they described as a new species, C. mahasarakhamense. The COI sequences obtained from the specimens morphologically identified as C. mahasarakhamense are similar/identical to 22 sequences previously reported by Jomkumsing et al. (2021) as C. arakawae. Therefore, Pramual et al. (2021) treated these 22 sequences as actually belonging to the new species, C. mahasarakhamense.

          For host blood meal analysis by Jomkumsing et al. (2021), 15 specimens from blood engorged females that had been morphologically identified as C. arakawae, indicated that they feed on chicken. Among these 15 specimens, 10 were later identified based on COI barcoding sequences as C. mahasarakhamense. Therefore, we concluded that C. mahasarakhamense is a chicken biter. Although we mentioned this, we did not intend to indicate that this species prefers to feed on chicken. Instead, we provided information that chicken is one of several possible blood meal sources for C. mahasarakhamense.

Reviewer 2 Report

Authors have prepared an excellent scientific manuscript.  Please note the following comments:

Line 50:  diseases are not transmitted.  The agents that cause diseases are transmitted. 

Last paragraph of Introduction:  Please more explicitly state the objective(s) of the study.

Line 99:  “Adult specimens”:  Are these all females?  The collecting procedure suggests that only females were collected.  Were all sequences from females?

Table 1:  To avoid confusion, please put more space between the column with Region and the column with N.  It looks like the Regions are Northeast 4, Northeast 6, Northeast10, etc.

Line 120 and Table 1:  Please explain if N = 86 sequences in the present study represents all individuals for which sequencing was attempted or if some individual midge sequencing efforts were not successful.

Lines 213-215:  If genetic divergence of the 3 haplotypes compared to the main lineage is considered “within the expected range of intraspecific genetic variations of Culicoides species”, are authors suggesting that all populations of C. mahasarakhamense represent a single species (but perhaps in an early stage of divergence)? 

Lines 259-260:  Because of the small size of the adults relative to their large surface area, the potential for dehydration is a serious problem. Therefore, humid conditions also would be advantageous for the adults. 

A careful reading by a native English language speaker will be helpful to correct English errors and making the manuscript more understandable.  For example, in the Abstract, please note:

Line 29:  “transmitted” (not “transmit”)

Line 40:  “that is genetically” (not “that genetically”)

Many more issues throughout the manuscript.

Author Response

Comments and Suggestions for Authors

Authors have prepared an excellent scientific manuscript.

Please note the following comments:

- Thank you very much for your comments and suggestions that are highly useful for manuscript improvement. We have revised the manuscript accordingly.

Line 50:  diseases are not transmitted.  The agents that cause diseases are transmitted.

- We have revised this to indicate that insect vectors are transmit “disease causing agents”.

Last paragraph of Introduction:  Please more explicitly state the objective(s) of the study.

- The objectives of the study were added.

Line 99:  “Adult specimens”:  Are these all females?  The collecting procedure suggests that only females were collected.  Were all sequences from females?

- Yes, all sequences were obtained from female specimens. We have modified the sentence to make it more specific.

Table 1:  To avoid confusion, please put more space between the column with Region and the column with N.  It looks like the Regions are Northeast 4, Northeast 6, Northeast10, etc.

- Space between the column with Region and N was added.

Line 120 and Table 1:  Please explain if N = 86 sequences in the present study represents all individuals for which sequencing was attempted or if some individual midge sequencing efforts were not successful.

- A total of 110 specimens were used for molecular analysis but only 86 were successful. This information was added to the Materials and Methods section.

Lines 213-215:  If genetic divergence of the 3 haplotypes compared to the main lineage is considered “within the expected range of intraspecific genetic variations of Culicoides species”, are authors suggesting that all populations of C. mahasarakhamense represent a single species (but perhaps in an early stage of divergence)? 

- Yes, we agree that it is possible that the divergent haplotypes could be representative of cryptic species or populations at an early stage of divergence. We have added discussion about this possibility.

Lines 259-260:  Because of the small size of the adults relative to their large surface area, the potential for dehydration is a serious problem. Therefore, humid conditions also would be advantageous for the adults. 

- Thank you for suggestion, we agree and have added this into the discussion.

A careful reading by a native English language speaker will be helpful to correct English errors and making the manuscript more understandable.  

For example, in the Abstract, please note:

Line 29:  “transmitted” (not “transmit”)

Line 40:  “that is genetically” (not “that genetically”)

Many more issues throughout the manuscript.

- Thank you very much for your suggestion. The manuscript was checked again by native English speaker.

Reviewer 3 Report

In this manuscript, genetic diversity, genetic structure and population history of the biting midge Culicoides mahasarakhamense has been revealed in Thailand. Presented data are clear and surely added new knowledge in this group of insect and and supported the important information for the effective disease control programs. However, in my opinion there are still some problems with this manuscript need to be improved.

Author Response

Comments and Suggestions for Authors

In this manuscript, genetic diversity, genetic structure and population history of the biting midge Culicoides mahasarakhamense has been revealed in Thailand. Presented data are clear and surely added new knowledge in this group of insect and and supported the important information for the effective disease control programs. However, in my opinion there are still some problems with this manuscript need to be improved.

- Thank you very much for your comments and suggestions that are helpful for manuscript improvement. We have revised manuscript accordingly.

I'm not sure what historical isolation means here?

- We have modified the sentence to make it clearer.

Biting midges is in duplication with the title, please.

- “Biting” was removed from the key words.

Line 60 – 71, If the focus of study is the population genetic study in the native range, then there is no need to elaborate too much about the genetic structure within newly colonized areas in the Introduction, and more attention should be paid to the research results related to the native range.

- The introduction section was revised according to reviewer comments and suggestions.

Line 73 – 74, Is this sentence mean that the insect is native from Thailand?

If your study is about the population from native range, then the above paragraph makes me confused.

- Culicoides mahasarakhamense is native to Thailand. We have modified the previous paragraph to make it clearer.

Figure 2, Mark the number of mutation steps in the figure

- The numbers of mutation steps (crossbars in the Figure 3) were added into figure 3.

Line 244 – 248, RE seems to be in the center of northeast region in the map.

- Yes, RE is located at the center of northeastern region. The interpretation that geographic distance is a possible factor responsible for genetic differentiation between RE and other populations is because those populations (SK, SR, UB, RB, LP) that are genetically significantly different from RE are separated by >120 km. Populations (MK1, MK2, MK3) that are not genetically significantly different are separate from RE by <100 km. An exception is LO which is separated from RE >300 km but they are not genetically significantly different. The explanation for this is small (n = 4) sample size for LO.

Round 2

Reviewer 1 Report

The authors have made an extensive effort for answering my concerns. I really appreciate the feedback. Therefore, the current version is improved, and I recommend to the editors for proceeding with its publication in Insects after considering five minor spelling issues.   

Line 60: Spelling check and full name, “Ades albopictus” by “Aedes albopictus Skuse

Line 69: Spelling check, “blue tongue” by “bluetongue virus”. https://ictv.global/

Line 74: This species is first mentioned here. “Culicoides brevitarsis Macfie”

Lines 91-92: Again, the diseases are not vectorized. I suggest “… or fully understanding the transmission ecology of these parasites that this biting midge species potential vector”

Lines 275: change “. If” by “, if”.

Author Response

Comments and Suggestions for Authors

The authors have made an extensive effort for answering my concerns. I really appreciate the feedback. Therefore, the current version is improved, and I recommend to the editors for proceeding with its publication in Insects after considering five minor spelling issues.   

- Thank you very much for your comments and suggestions. All are very helpful for improvement of the manuscript. We are really appreciating for this.

Line 60: Spelling check and full name, “Ades albopictus” by “Aedes albopictus Skuse

- Corrected.

Line 69: Spelling check, “blue tongue” by “bluetongue virus”. https://ictv.global/

- Corrected.

Line 74: This species is first mentioned here. “Culicoides brevitarsis Macfie”

- Corrected.

Lines 91-92: Again, the diseases are not vectorized. I suggest “… or fully understanding the transmission ecology of these parasites that this biting midge species potential vector”

- Corrected.

Lines 275: change “. If” by “, if”.

- Corrected.

Reviewer 3 Report

agree to accept the article

Author Response

Comments and Suggestions for Authors

agree to accept the article

- Thank you very much. We really appreciate for your comments and suggestions for improvement of the manuscript.